# Genetic timestamping of plasma cells in vivo reveals tissue-specific homeostatic population turnover

An Qi Xu[1], Rita R Barbosa[1], Dinis Pedro Calado[1,2]*

[1]Immunity and Cancer, The Francis Crick Institute, London, United Kingdom; [2]Peter Gorer Department of Immunobiology, School of Immunology & Microbial Sciences, King's College London, London, United Kingdom

**Abstract** Plasma cells (PCs) are essential for protection from infection, and at the origin of incurable cancers. Current studies do not circumvent the limitations of removing PCs from their microenvironment and confound formation and maintenance. Also, the investigation of PC population dynamics has mostly relied on nucleotide analog incorporation that does not label quiescent cells, a property of most PCs. The main impediment is the lack of tools to perform specific genetic manipulation in vivo. Here we characterize a genetic tool (Jchain^creERT2) in the mouse that permits first-ever specific genetic manipulation in PCs in vivo, across immunoglobulin isotypes. Using this tool, we found that splenic and bone marrow PC numbers remained constant over-time with the decay in genetically labeled PCs being compensated by unlabeled PCs, supporting homeostatic population turnover in these tissues. The Jchain^creERT2 tool paves the way for an in-depth mechanistic understanding of PC biology and pathology in vivo, in their microenvironment.

## Introduction

Antibodies produced by plasma cells (PCs) are crucial for immune protection against infection and for vaccination success (*Nutt et al., 2015*). Upon activation, B cells terminally differentiate into PCs, a process initiated by the downregulation of the B cell transcription factor PAX5 (*Kallies et al., 2007*). This event allows the expression of multiple factors normally repressed by PAX5, including *Xbp1* and *Jchain* (*Castro and Flajnik, 2014*; *Nutt et al., 2015*; *Rinkenberger et al., 1996*; *Shaffer et al., 2004*). PAX5 downregulation is also followed by the expression of the transcription factors IRF4 and BLIMP1 that play essential roles in the establishment of the PC program (*Kallies et al., 2007*; *Klein et al., 2006*; *Sciammas et al., 2006*; *Shapiro-Shelef et al., 2003*). Beyond physiology, multiple cancers have a PC as cell of origin, including multiple myeloma, the second most frequent hematological malignancy overall, and for which a cure remains to be found (*Palumbo and Anderson, 2011*). As a consequence, the study of gene function in PC biology and pathology is a subject of intense investigation.

However, at least in part because of technical limitations most PC studies make use of in vitro and cell transfer systems that remove PCs from their microenvironment. Currently, genetic manipulation of PCs is not specific and targets other cell populations such as B cells, confounding PC formation, and maintenance. Also, studies on the turnover of the PC population are lacking, as investigation of the regulation of PC maintenance has mostly relied on the use of nucleotide analogs that do not track the vast majority of PCs due to their quiescent nature.

We found amongst well-known PC-associated genes, that *Jchain* (*Igj*) had the highest level and most specific expression in PC populations. JCHAIN is a small polypeptide required to multimerize IgM and IgA, and necessary for the transport of these Ig classes across the mucosal epithelium in a

*For correspondence:
dinis.calado@crick.ac.uk

Competing interests: The authors declare that no competing interests exist.

poly-Ig receptor-mediated process (*Brandtzaeg and Prydz, 1984*; *Castro and Flajnik, 2014*; *Max and Korsmeyer, 1985*). Here we characterized in detail a GFP-tagged creERT2 allele at the *Jchain* endogenous locus: *Igj*<sup>creERT2</sup>, hereafter termed *Jchain*<sup>creERT2</sup>. We found at the single cell level that GFP as a reporter of *Jchain* expression occurred in PCs across immunoglobulin isotypes, including IgG1. Using the *Jchain*<sup>creERT2</sup> allele we performed the first-ever highly specific cre-loxP genetic manipulation in PCs residing in their natural microenvironment in vivo. This system allowed inclusive genetic timestamping of PCs, independently of their cell cycle status. We uncovered that the number of PCs in the spleen and bone marrow remained constant over-time with the decay in numbers of genetically labeled PCs being compensated by that of unlabeled PCs, supporting homeostatic population turnover in these tissues. The *Jchain*<sup>creERT2</sup> is thus a validated PC specific genetic tool that paves the way for an in-depth mechanistic understanding of PC biology and pathology in vivo, in their microenvironment.

## Results

### *Jchain* transcripts are highly enriched in plasma cells

B-to-PC differentiation is a process that involves a complex network of factors (*Figure 1A*; *Nutt et al., 2015*). We investigated the level and specificity of the expression of genes associated with PCs (*Xbp1*, *Jchain*, *Scd1*, *Irf4*, and *Prdm1*) through the analysis of a publicly available RNA sequencing dataset for immune cell populations (ImmGen, [*Heng et al., 2008*]). We first determined the cell populations with the highest transcript level for each factor. *Xbp1* and *Irf4* were primarily expressed in PCs, however, the expression in bone marrow PCs (B_PC_BM) was less than two-fold greater than that of non-PC populations (*Figure 1B*). The expression of *Sdc1* and *Prdm1* was not specific to PCs (*Figure 1C*). Notably, peritoneal cavity macrophages (MF_226+II+480lo_PC) expressed more *Sdc1* than bone marrow PCs (B_PC_BM), and a subset of FOXP3<sup>+</sup> T cells (Treg_4_FP3+_Nrplo_Co) expressed higher levels of *Prdm1* than that observed in splenic plasmablasts (B_PB_Sp) and bone marrow PCs (B_PC_BM; *Figure 1C*). By contrast, *Jchain* had the highest level of transcript expression in PCs compared to non-PCs and was the most PC specific amongst all factors, with a forty-fold enrichment over germinal center (GC) B cells (B_GC_CB_Sp; *Figure 1D*). We concluded that the *Jchain* locus was a suitable candidate for the generation of PC specific genetic tools.

### *Jchain* is expressed in a small fraction of GC B cells and in most plasma cells

We searched alleles produced by the EUCOMMTools consortium (*Koscielny et al., 2014*) and identified a genetically engineered *Jchain* allele produced by the Wellcome Trust Sanger Institute: MGI:5633773, hereafter termed *Jchain*<sup>creERT2</sup>. The genetically engineered *Jchain* allele contained an *FRT* site between exons 1 and 2 followed by an engrailed two splice acceptor sequence and an *EGFP.2A.cre*<sup>ERT2</sup> expression cassette (*Figure 2A*). In this design, the expression of the EGFP (GFP) and of cre<sup>ERT2</sup> is linked by a self-cleaving 2A peptide under the transcriptional control of the *Jchain* promoter (*Figure 2A*). To determine cells with GFP expression, we initially analyzed the spleen from mice heterozygous for the *Jchain*<sup>creERT2</sup> allele that had been immunized with sheep red blood cells (SRBC) 12 days earlier (*Figure 2B*). B220 is expressed on the surface of B cells and downregulated during PC differentiation (*Pracht et al., 2017*). We, therefore, defined three cell populations based on the levels of GFP fluorescence and B220 surface expression: GFP<sup>low</sup>B220<sup>high</sup>, GFP<sup>int</sup>B220<sup>int</sup>, GFP<sup>high</sup>B220<sup>low</sup>, and a population negative for both markers (GFP<sup>neg</sup>B220<sup>neg</sup>; *Figure 2C*). Next, we determined the fraction of cells within these populations that expressed surface CD138, a commonly used marker to define PCs by flow-cytometry (*Pracht et al., 2017*). The GFP<sup>neg</sup>B220<sup>neg</sup> population did not contain CD138<sup>+</sup> cells, however, the fraction of CD138<sup>+</sup> cells increased in the remaining populations in agreement with the reduction of B220 expression during PC differentiation, and the GFP<sup>int</sup>B220<sup>int</sup> and GFP<sup>high</sup>B220<sup>low</sup> populations were mostly composed of CD138<sup>+</sup> cells (*Figure 2D, E*). Identical results were found when defining PCs using in addition surface expression of CXCR4, a chemokine receptor that facilitates homing of PCs to the bone marrow (*Figure 2D,E*; *Hargreaves et al., 2001*). Thus, increased GFP expression from the *Jchain*<sup>creERT2</sup> allele associates with the loss of B220 and increased expression of PC-associated surface markers.

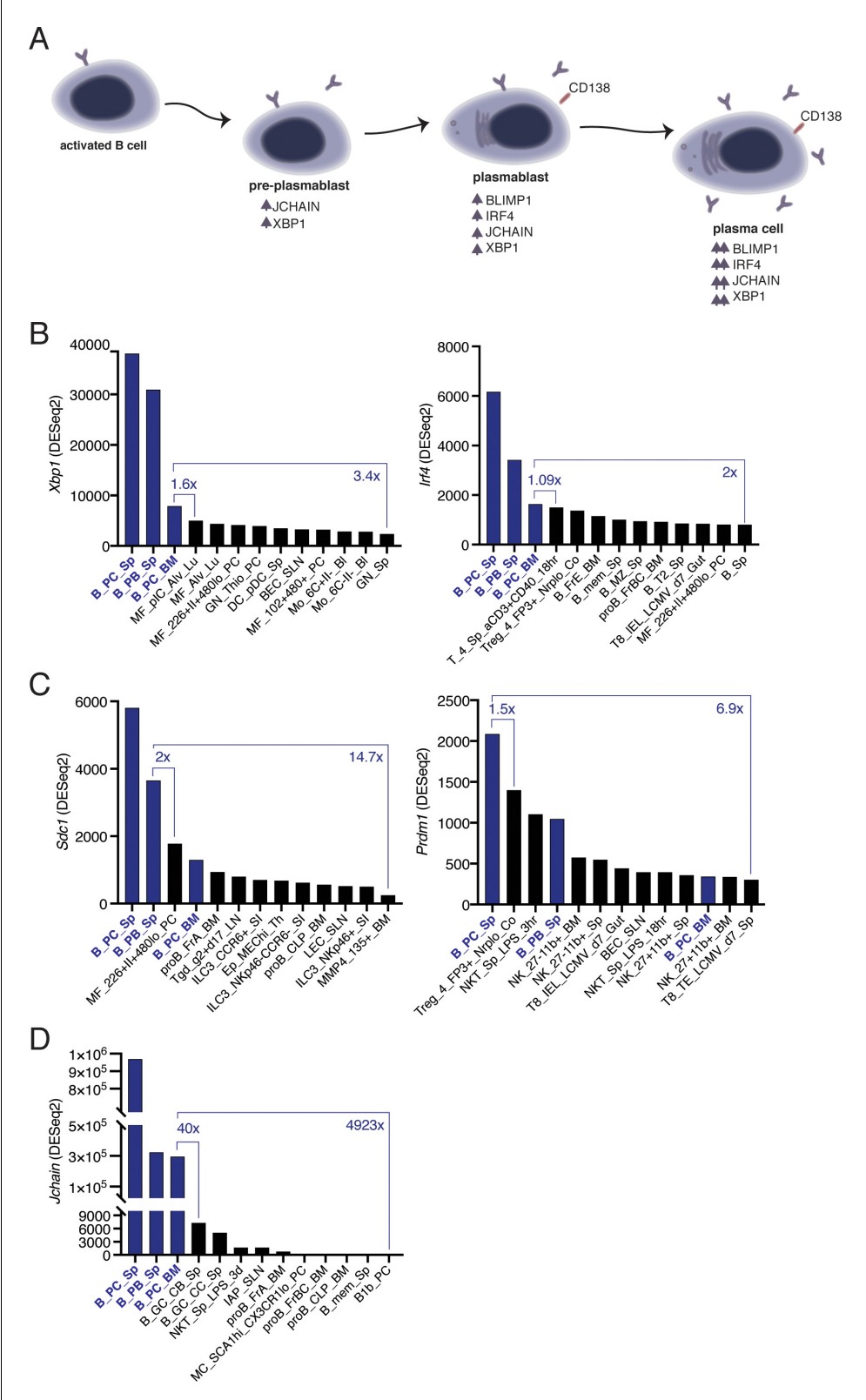

**Figure 1.** *Jchain* transcripts are highly enriched in plasma cells. (**A**) Schematic of the network of factors associated with plasma cell differentiation. Upward arrows indicate increased expression compared to the precursor population. (**B–D**) Differential gene expression analysis using RNA sequencing data (ImmGen, *Heng et al., 2008*). (**B**) Analysis of *Xbp1* and *Irf4*, which encode for XBP1 and IRF4, respectively. (**C**) Analysis of *Sdc1* and *Prdm1*, which

*Figure 1 continued on next page*

*Figure 1 continued*

encode for CD138 and BLIMP1, respectively. (D) Analysis for *Jchain* (*Igj*) that encodes for JCHAIN. 'x' indicates fold change. Expression Value Normalized by DESeq2. http://rstats.immgen.org/Skyline/skyline.html.

The online version of this article includes the following source data for figure 1:

**Source data 1.** Gene expression values normalised by DESeq2.

We further investigated the cellular composition of the GFP$^{low}$B220$^{high}$ population that contained the fewest CD138$^+$ cells (1% to 20%; *Figure 2E*). The gate defining this population was designed to not disregard the occurrence of cells with a low-level of GFP expression in mice carrying the *Jchain*$^{creERT2}$ allele. However, such strategy inevitably led to the inclusion of a small fraction of 'false' GFP positive cells as indicated by the analysis of control mice (wild-type C57BL/6) that are GFP negative (*Figure 2C*). Still, the fraction of cells within the GFP$^{lo}$B220$^{hi}$ population was significantly enriched in *Jchain*$^{creERT2}$ mice compared to control (*Figure 2—figure supplement 1*). Also above background, we found that virtually all CD138$^{neg}$ cells within the GFP$^{low}$B220$^{high}$ population of mice with the *Jchain*$^{creERT2}$ allele expressed the B cell marker CD19 (*Figure 2F*), and in agreement with *Jchain* gene expression analysis (*Figure 1D*) these cells mostly represented germinal center (GC) B cells (CD38$^{low}$FAS$^{high}$; *Figure 2F,G* and *Figure 2—figure supplement 1*). By contrast, only 30% of *Jchain*$^{creERT2}$ mice had above background enrichment for CD138$^+$ and CD138$^+$CXCR4$^+$ expressing cells within the GFP$^{low}$B220$^{high}$ population (*Figure 2—figure supplement 1*). These data prompts caution when using the gate defining the GFP$^{low}$B220$^{high}$ population to study enrichment for PC markers in mice carrying *Jchain*$^{creERT2}$ allele, as it may contain an unacceptable level of contamination by non-GFP positive cells.

We further performed analyses in the spleen and bone marrow of 12 day SRBC immunized mice heterozygous for the *Jchain*$^{creERT2}$ allele in which precursors and mature B cells, and non-B cell populations were first defined using surface markers and the fraction of GFP expressing cells within those populations determined (*Figure 2—figure supplement 2*). We found that most PCs in the spleen and bone marrow expressed GFP (CD138$^+$CXCR4$^+$, 60 to 90%; *Figure 2—figure supplement 2*). We also observed that a minor fraction of B1b cells (0 to 6%) in the spleen expressed GFP (*Figure 2—figure supplement 2*), possibly in agreement with the knowledge that B1b cells are prone to differentiate into PCs and are a source of IgM antibodies during T cell independent responses (*Alugupalli et al., 2004*). Collectively these data confirmed at the single cell level the gene expression analysis using bulk populations (*Figure 1D*) and suggested that *Jchain* expression is highly enriched in PCs.

### *Jchain* expression correlates with that of IRF4 and BLIMP1

IRF4 and BLIMP1 transcription factors play an essential role in PC differentiation (*Kallies et al., 2007*; *Klein et al., 2006*; *Sciammas et al., 2006*; *Shapiro-Shelef et al., 2003*). We analyzed the spleen of 12 day SRBC immunized mice heterozygous for the *Jchain*$^{creERT2}$ allele and determined the expression pattern of IRF4 and BLIMP1 in the populations defined by varied GFP and B220 expression (*Figures 2C* and *3A,B*). The GFP$^{neg}$B220$^{neg}$ population was virtually devoid of cells with BLIMP1 and IRF4 expression (*Figure 3C*). In 30% of mice carrying the *Jchain*$^{creERT2}$ allele we found above background enrichment for BLIMP1$^+$IRF4$^+$ cells within the GFP$^{low}$B220$^{high}$ population (*Figure 3D,E* and *Figure 3—figure supplement 1*). However, most GFP$^{low}$B220$^{high}$ cells were negative for BLIMP1 and IRF4, suggesting that *Jchain* expression precedes that of IRF4 and BLIMP1, as previously observed in in vitro cultures of *Blimp1* deficient B cells (*Kallies et al., 2007*). Still, *Jchain* expression as measured by GFP strongly correlated with that of IRF4 and BLIMP1 given that the vast majority of cells within the GFP$^{int}$B220$^{int}$ and GFP$^{high}$B220$^{low}$ populations were BLIMP1$^+$IRF4$^+$ (*Figure 3D and E*). Overall, we identified an in vivo population of cells in which *Jchain* expression preceded that of IRF4 and BLIMP1, possibly representing PC precursors. As PC differentiation ensued, *Jchain* expression correlated highly with the expression of the transcription factors BLIMP1 and IRF4 that are critical for the establishment of the PC program.

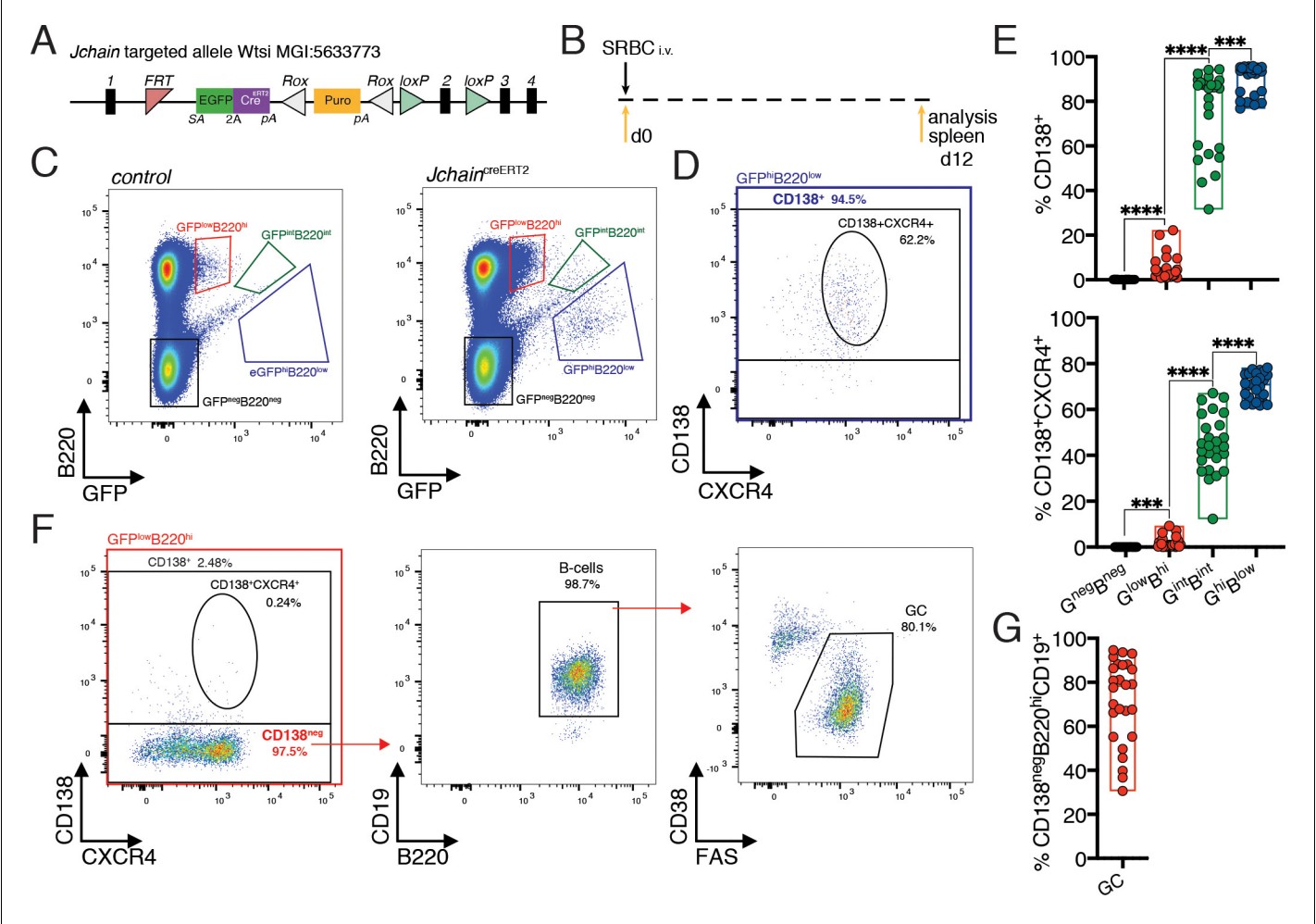

**Figure 2.** *Jchain* is expressed in a small fraction of GC B cells and in most plasma cells. (**A**) Schematic of *Jchain* targeted allele Wtsi MGI:5633773. Rectangular boxes indicate exons, and exon number is on top; pink triangle indicates an FRT sequence; EGFPcre[ERT2] cassette contains a splice acceptor site (SA)-led EGFP-2A-creERT2 expression cassette followed by a poly-A tail inserted in the intron between exons 1 and 2; white triangle indicates a ROX sequence; orange rectangle indicates a promoter-driven puromycin resistance cassette; green triangle indicates loxP sequence. (**B**) Schematic of experimental procedure protocol. Mice carrying the *Jchain*[creERT2] allele were immunized with sheep red blood cells (SRBC) intravenously (i.v.) on day 0 and spleens of mice were analyzed at day 12 post-immunization. (**C**) Gating strategy of populations by flow-cytometry according to the expression of GFP and B220 in mice carrying the *Jchain*[creERT2] allele and wild-type B6 mice for a negative control of GFP expression. (**D**) Gating strategy by flow-cytometry for plasma cells within the GFP[hi]B220[low] population using CD138[+] and CD138[+]CXCR4[+] markers. (**E**) Cumulative data for CD138[+] and CD138[+]CXCR4[+]plasma cells analyzed as in (**D**). Top: fraction of CD138[+]plasma cells; bottom: fraction of CD138[+]CXCR4[+]plasma cells within the four populations defined by flow-cytometry according to the expression of GFP and B220 in mice carrying the *Jchain*[creERT2] allele. (**F**) Gating strategy by flow-cytometry for total CD138[+] and CD138[+]CXCR4[+]plasma cells within GFP[low]B220[hi] population. The CD138[neg] cell fraction within the GFP[low]B220[hi] population was analyzed for the CD19 B cell marker and stained for CD38 and FAS to determine germinal center (GC) B cells. (**G**) Cumulative data for the frequency of GC B cells within the CD138[neg]GFP[low]B220[hi] population. Each symbol (E: n = 26, G: n = 26) represents an individual mouse; small horizontal lines indicate median, minimum, and maximum values. ***=p ≤ 0.001, ****=p ≤ 0.0001 (unpaired Student's *t*-test). Data are representative of three independent experiments (**E, G**).

The online version of this article includes the following source data and figure supplement(s) for figure 2:

**Source data 1.** Frequency of plasma cells or B cells within the populations defined by GFP and B220 expression.

**Figure supplement 1.** Cell marker enrichment within the GFP[low]B220[hi] population.

**Figure supplement 1—source data 1.** Cell marker enrichment within the GFP[low]B220[hi] population.

**Figure supplement 2.** *Jchain* expression in multiple stages of B cell development and other cell lineages.

**Figure supplement 2—source data 1.** Frequency of GFP[+] cells within the various immune populations in the spleen and bone marrow.

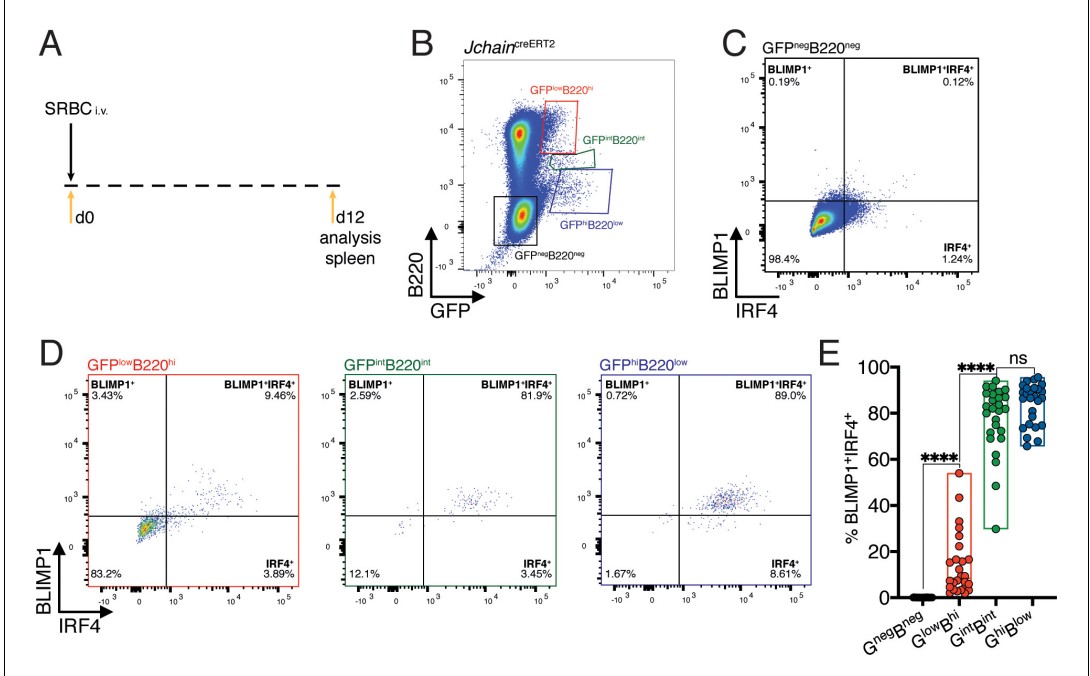

**Figure 3.** *Jchain* expression correlates with that of IRF4 and BLIMP1. (**A**) Schematic of experimental procedure protocol. Mice carrying the *Jchain*^creERT2 allele were immunized with sheep red blood cells (SRBC) intravenously (i.v.) on day 0 and spleens of mice were analyzed at day 12 post-immunization. (**B**) Gating strategy of populations by flow-cytometry according to the expression of GFP and B220 in mice carrying the *Jchain*^creERT2 allele. (**C**) Gating strategy for IRF4 and BLIMP1 expression by flow-cytometry within the GFP^neg B220^neg population. (**D**) Gating strategy for IRF4 and BLIMP1 expression by flow-cytometry within GFP^low B220^hi, GFP^int B220^int, and GFP^hi B220^low populations, defined as in (**B**). (**E**) Cumulative data for the frequency of BLIMP1^+IRF4^+ cells within the four populations defined as in (**B**). Each symbol in (E: n = 26) represents an individual mouse; small horizontal lines indicate median, minimum, and maximum values. ns = not significant, ***=$p \leq 0.001$, ****=$p \leq 0.0001$ (unpaired Student's *t*-test). Data are representative of three independent experiments (**E**).

The online version of this article includes the following source data and figure supplement(s) for figure 3:

**Source data 1.** Frequency of BLIMP1^+IRF4^+ cells within the populations defined by GFP and B220 expression.

**Figure supplement 1.** BLIMP1^+IRF4^+ cells within the GFP^low B220^hi population.

**Figure supplement 1—source data 1.** Frequency of BLIMP1^+IRF4^+ cells within the GFP^low B220^hi population.

## *Jchain*^creERT2 mediated genetic manipulation is effective only in plasma cells

Next, we sought to determine whether the *Jchain*^creERT2 allele could be used to perform genetic manipulation in PCs. For that we generated compound mutant mice carrying the *Jchain*^creERT2 allele and a Rosa 26 allele in which RFP expression is conditional to cre-mediated recombination of a loxP-STOP-loxP cassette (*R26*^lslRFP; *Luche et al., 2007*). In the *Jchain*^creERT2 allele, cre is fused to an estrogen binding domain (ERT2) that sequesters cre in the cytoplasm through the binding to HSP90 (*Feil et al., 2009*). Addition of tamoxifen displaces the creERT2-HSP90 complex allowing effective nuclear import of creERT2 and its access to loxP flanked DNA sequences (*Figure 4A*; *Feil et al., 2009*). We first performed an in vitro experiment using a classical plasmablast (B220^low CD138^+) differentiation assay in which B cells purified from mice carrying the *Jchain*^creERT2 and *R26*^lslRFP alleles were cultured with LPS (*Andersson et al., 1972*) in the presence or absence of 4-OH tamoxifen. GFP expression was highly enriched in plasmablasts compared to B cells and RFP expression was only observed upon addition of 4-OH tamoxifen to the cell culture, and that occurred virtually only in plasmablasts (*Figure 4—figure supplement 1*). This data suggested that creERT2 was specifically expressed by plasmablasts and effectively retained in the cytoplasm in the absence of 4-OH tamoxifen.

Similar observations were made in mice carrying the *Jchain*^creERT2 and *R26*^lslRFP alleles in vivo. In the absence of tamoxifen administration RFP expressing cells were not detected, confirming the in

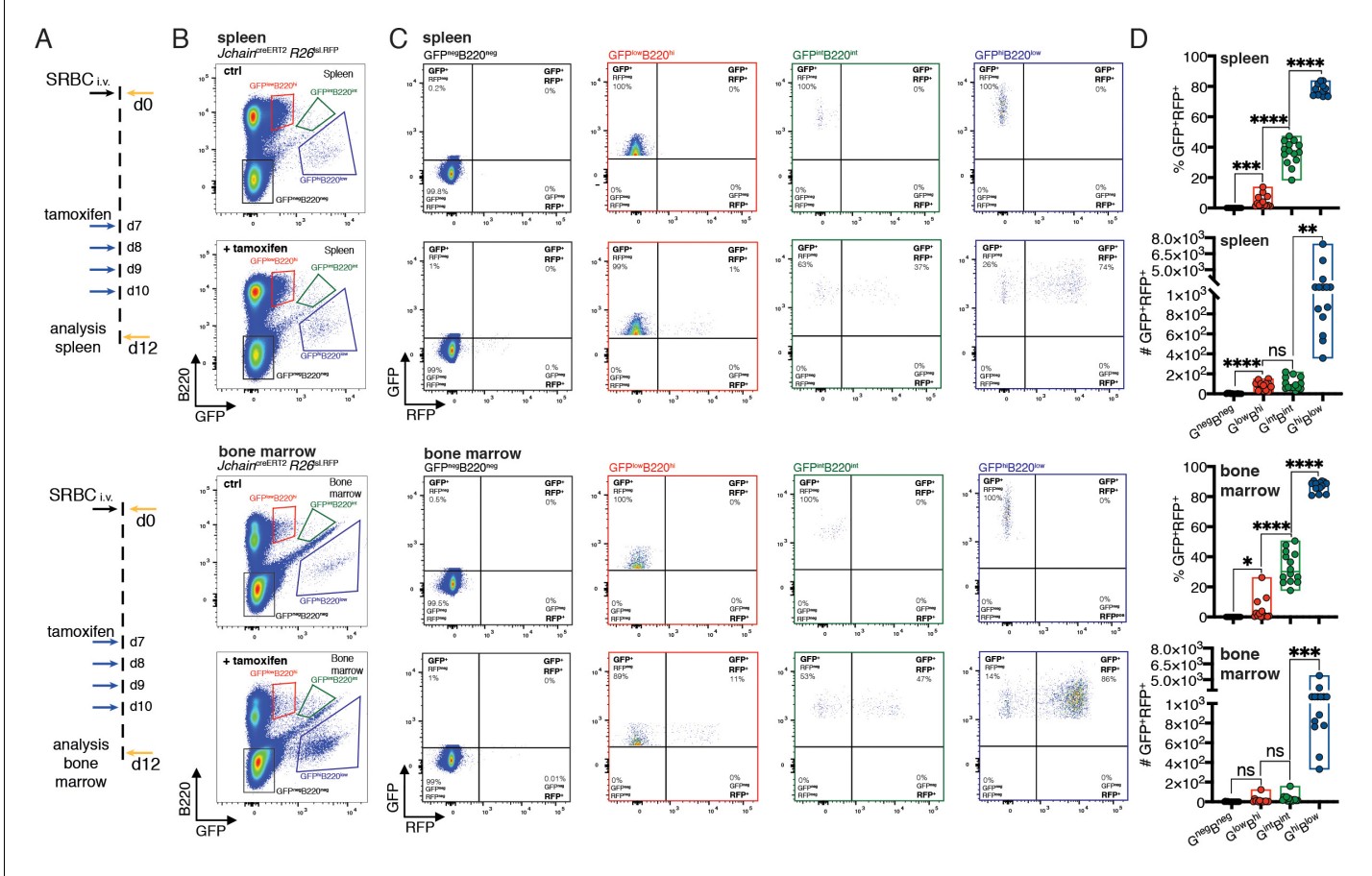

**Figure 4.** *Jchain*$^{creERT2}$ mediated genetic manipulation is effective only in plasma cells. (**A**) Schematic of experimental procedure protocol. Mice carrying the *Jchain*$^{creERT2}$ and *R26*$^{lslRFP}$ alleles were immunized with sheep red blood cells (SRBC) intravenously (i.v.) on day 0 and spleens (top) and bone marrow (bottom) of mice were analyzed at day 12 post-immunization. A group of mice received tamoxifen treatment for four consecutive days from day 7 to day 10 after immunization. (**B**) Gating strategy of populations by flow-cytometry in the spleen (top) and bone marrow (bottom) according to the expression of GFP and B220 in mice carrying the *Jchain*$^{creERT2}$ allele. (**C**) Gating strategy for GFP and RFP expression by flow-cytometry in the four populations defined as in (**B**). Top: spleen; bottom: bone marrow. (**D**) Cumulative data for the frequency and number of GFP$^+$RFP$^+$ cells within GFP$^{neg}$B220$^{neg}$, GFP$^{low}$B220$^{hi}$, GFP$^{int}$B220$^{int}$, and GFP$^{hi}$B220$^{low}$ populations, defined as in (**B**). Top: spleen; bottom: bone marrow. Each symbol (D: n = 14) represents an individual mouse; small horizontal lines indicate median, minimum, and maximum values. ns = not significant, *=p $\leq$ 0.05, **=p $\leq$ 0.01, ***=p $\leq$ 0.001, ****=p $\leq$ 0.0001 (unpaired Student's *t*-test). Data are representative of three independent experiments (**D**).

The online version of this article includes the following source data and figure supplement(s) for figure 4:

**Source data 1.** Frequency and number of GFP$^+$RFP$^+$ cells within the populations defined by GFP and B220 expression.

**Figure supplement 1.** *Jchain*$^{creERT2}$ mediated genetic manipulation in vitro.

**Figure supplement 1—source data 1.** Fractions of populations defined by GFP and RFP expression within B cells or plasmablasts in vitro.

**Figure supplement 2.** The cre activity of the *Jchain*$^{creERT2}$ allele is tightly regulated.

**Figure supplement 2—source data 1.** Frequency of GFP$^+$RFP$^+$ cells within the various immune populations in the spleen and bone marrow.

vitro results and supporting that the *Jchain*$^{creERT2}$ allele was not 'leaky' in the control of cre activity (*Figure 4—figure supplement 2*). Next, we immunized mice carrying *Jchain*$^{creERT2}$ and *R26*$^{lslRFP}$ alleles and administered tamoxifen on days 7, 8, 9, and 10 followed by analysis of the spleen and bone marrow at day 12 (*Figure 4A*). Analyses of the populations defined by varied GFP and B220 surface expression (*Figures 2* and *4B*) revealed that a small fraction of GFP$^{low}$B220$^{high}$ cells were positive for RFP in the spleen (median ~2%) and in the bone marrow (median ~1.2%; *Figure 4C,D*). In contrast, cre mediated recombination and as consequence RFP expression occurred in ~37% and ~31% of GFP$^{int}$B220$^{int}$ in the spleen and bone marrow, respectively, and the vast majority of cells within the GFP$^{high}$B220$^{low}$ population had undergone cre mediated recombination and were RFP positive (~76% in the spleen, and ~88% in the bone marrow, median; *Figure 4C,D*). These data

showed that *Jchain*<sup>creERT2</sup> mediated cre-recombination was only effective in PC populations validating it as a tool to specifically perform genetic manipulation of PCs.

## Genetic manipulation using *Jchain*<sup>creERT2</sup> occurs across immunoglobulin isotypes

IgG1 does not multimerize, and due to differences in its secretory tail to that of IgA and IgM, JCHAIN does not associate with IgG1 (*Johansen et al., 2000*). Currently it is suggested that *Jchain* expression occurs in all PCs regardless of isotype (*Castro and Flajnik, 2014*; *Johansen et al., 2000*; *Mather et al., 1981*). However, this has not been demonstrated at the single cell level. We performed experiments that investigated whether *Jchain*<sup>creERT2</sup> allele GFP expression and cre-mediated loxP recombination occurred in PCs across immunoglobulin isotypes. For that we analyzed the spleen, mesenteric lymph node (mLN), Peyer's patches and bone marrow of younger (15 weeks) and older (30 weeks) mice carrying the *Jchain*<sup>creERT2</sup> and *R26*<sup>lslRFP</sup> alleles. These mice were immunized with SRBC and administered with tamoxifen on days 7, 8, 9, and 10 followed by analysis at day 12 (*Figure 5A*). We first analyzed total PCs (B220<sup>low</sup>CD138<sup>+</sup>), and within these cells those that expressed GFP (RFP<sup>+</sup> and RFP<sup>neg</sup>) and GFP<sup>+</sup>RFP<sup>+</sup> cells (cre recombined) to determine the proportions of IgA, IgM, and IgG1 expressing cells using intracellular stain (*Figure 5B–D*). Overall, we found only small differences. Analysis of spleens of 15-week-old mice revealed a slight increase in the percentage of IgA<sup>+</sup> cells within the GFP<sup>+</sup>RFP<sup>+</sup> PCs only when compared to total PCs (*Figure 5E*). A similar trend was observed when analyzing Peyer's patches of 15-week-old mice (*Figure 5E*). However, in the other analyzed tissues of these mice we did not observe differences in the proportion of IgA, nor for IgM and IgG1 in any of the tissues analyzed (*Figure 5E*). 30-week-old mice showed a slight increase in the percentage of splenic IgM<sup>+</sup> cells within the GFP<sup>+</sup>RFP<sup>+</sup> PCs only when compared to total PCs, and for IgA<sup>+</sup> in the mLN (*Figure 5F*). No significant difference was observed in 30-week-old mice for the proportion of IgM or IgA in any of the other analyzed tissues, and in none of the analyzed tissues for IgG1 (*Figure 5F*). Taken together, these data suggested that *Jchain* expression is not overly represented in IgA or IgM expressing PCs compared to IgG1<sup>+</sup> PCs. We concluded that *Jchain*<sup>creERT2</sup> mediated cre-loxP recombination occurs across immunoglobulin isotypes and thus that the *Jchain*<sup>creERT2</sup> allele is a useful tool for genetic manipulation also of IgG1 expressing PCs.

## Inclusive analysis of plasma cell dynamics reveals tissue-specific homeostatic population turnover

Understanding of the PC population turnover is lacking. Multiple investigations have been performed to determine the PC lifespan using primarily nucleotide analog incorporation into the DNA. These studies have provided fundamental insights on PC maintenance and currently it is accepted that a fraction of PCs in the mouse survives for periods longer than 3 months (*Ho et al., 1986*; *Lemke et al., 2016*; *Manz et al., 1998*; *Manz et al., 1997*; *Slifka et al., 1998*). However, given the quiescent nature of PCs and that nucleotide analog methodology requires cell division, these methods are not appropriate to study global population turnover in tissues. We investigated the suitability of the *Jchain*<sup>creERT2</sup> allele to determine the turnover of the PC population in the spleen and bone marrow. We immunized mice carrying the *Jchain*<sup>creERT2</sup> and *R26*<sup>lslRFP</sup> alleles at two time-points spaced by a period of 21 days (*Figure 6A*; *Calado et al., 2010*). Thirty days after the secondary immunization (day 51) we administered tamoxifen for five consecutive days to genetically label PCs (*Figure 6A*). Next, we determined the absolute cell number of total PCs, of GFP<sup>+</sup> cells (RFP<sup>+</sup> and RFP<sup>neg</sup>), GFP<sup>+</sup>RFP<sup>+</sup> cells (cre recombined), and GFP<sup>+</sup>RFP<sup>neg</sup> cells (not cre recombined). We found that over a period of 5 months the cell number of total and GFP<sup>+</sup> PCs remained constant over time (*Figure 6B,C*). By contrast, the cell number of GFP<sup>+</sup>RFP<sup>+</sup> cells decayed (*Figure 6B,C*) in both the spleen ($t_{1/2}$ ~31.63d) and bone marrow ($t_{1/2}$ ~251.93d; *Figure 6D,E*). These results may agree with the knowledge that the half-life of PC residence differs between spleen and the bone marrow (*Sze et al., 2000*). Notably, analysis of the GFP<sup>+</sup>RFP<sup>neg</sup> cell numbers revealed that the emergence of these cells paralleled that observed for the decay in cell numbers of GFP<sup>+</sup>RFP<sup>+</sup> cells (*Figure 6B,C*) both in the spleen ($t_{1/2}$ ~20.20d) and bone marrow ($t_{1/2}$ ~190.19d; *Figure 6F,G*). These data indicated that the turnover of the PC population is homeostatically regulated in a tissue-specific manner.

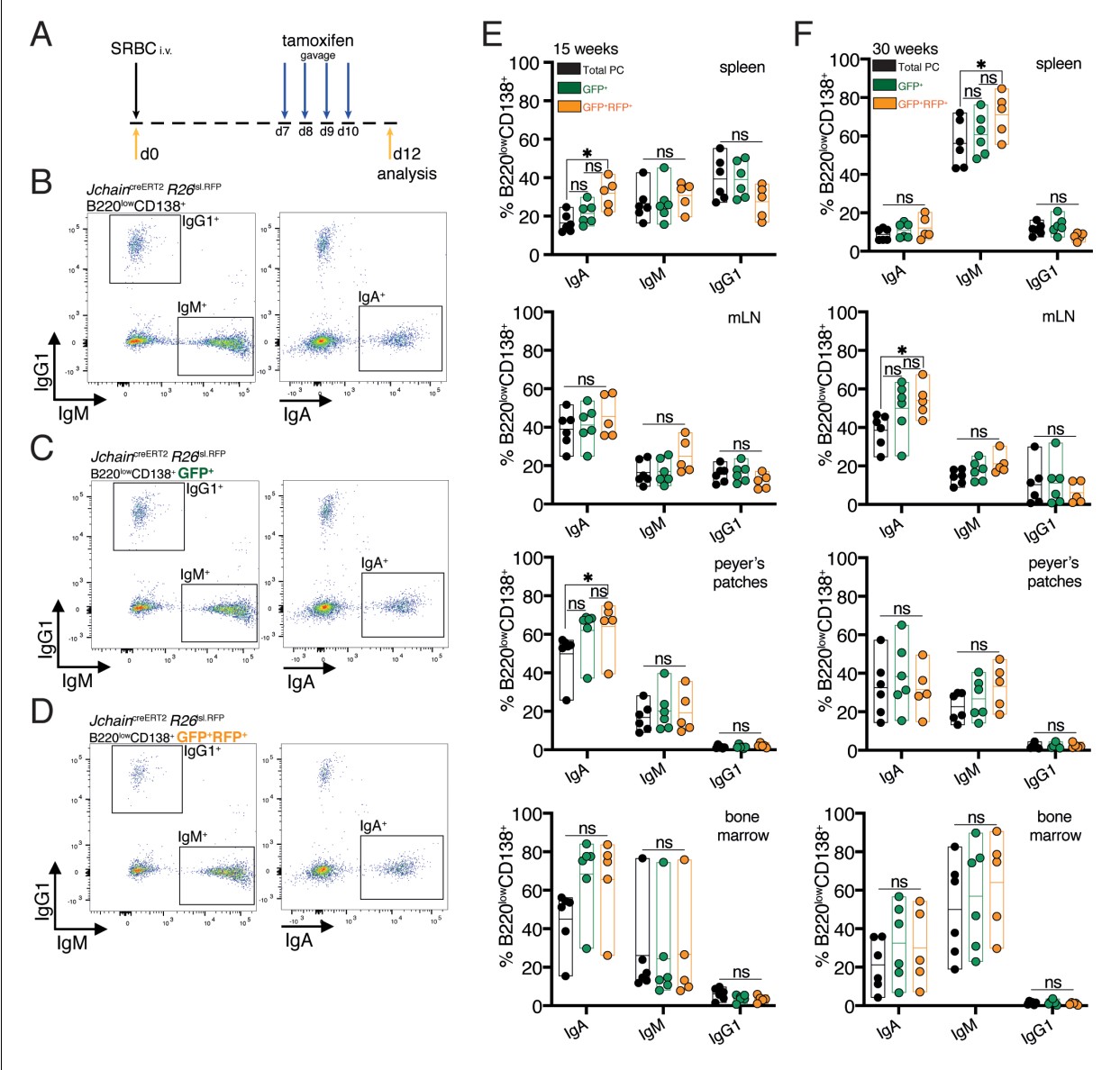

**Figure 5.** Genetic manipulation using *Jchain*$^{creERT2}$ occurs across immunoglobulin isotypes. (**A**) Schematic of experimental procedure protocol. Mice carrying the *Jchain*$^{creERT2}$ and *R26*$^{lslRFP}$ alleles were immunized with sheep red blood cells (SRBC) intravenously (i.v.) on day 0 and spleens, mesenteric lymph nodes (mLN), Peyer's patches, and bone marrows of mice were analyzed at day 12 post-immunization. Mice received tamoxifen treatment for four consecutive days from day 7 to day 10 after immunization. (**B**) Gating strategy by flow-cytometry for intracellular and extracellular expression of IgG1, IgM and IgA within total B220$^{low}$CD138$^+$plasma cells. Analysis in the spleen is provided as example. (**C**) Gating strategy by flow-cytometry for intracellular and extracellular expression of IgG1, IgM, and IgA within B220$^{low}$CD138$^+$GFP$^+$ (RFP$^+$ and RFP$^{neg}$) plasma cells. Analysis in the spleen is provided as example. (**D**) Gating strategy by flow-cytometry for intracellular and extracellular expression of IgG1, IgM, and IgA within B220$^{low}$CD138$^+$GFP$^+$RFP$^+$plasma cells. Analysis in the spleen is provided as example. (**E**) Cumulative data for the fractions of IgA, IgM or IgG1 expressing cells within total plasma cells (PC) (black, B220$^{low}$CD138$^+$), GFP$^+$ (RFP$^+$ and RFP$^{neg}$) plasma cells (green, B220$^{low}$CD138$^+$GFP$^+$), and GFP$^+$RFP$^+$plasma cells (orange, B220$^{low}$CD138$^+$GFP$^+$RFP$^+$) at 15 weeks of age. (**F**) Cumulative data for the fractions of IgA, IgM, or IgG1 expressing cells within total plasma cells (PC) (black, B220$^{low}$CD138$^+$), GFP$^+$ (RFP$^+$ and RFP$^{neg}$) plasma cells (green, B220$^{low}$CD138$^+$GFP$^+$), and GFP$^+$RFP$^+$plasma cells (orange, B220$^{low}$CD138$^+$GFP$^+$RFP$^+$) at 30 weeks of age. Each symbol (E: n = 5–6, F: n = 5–6) represents an individual mouse; small horizontal lines indicate median, minimum, and maximum values. ns = not significant, *=p ≤ 0.05 (unpaired Student's *t*-test). Data are representative of three independent experiments (**E**, **F**).

The online version of this article includes the following source data for figure 5:

**Source data 1.** Fractions of IgA, IgM or IgG1-expressing cells within populations of plasma cells defined by GFP and RFP expression at various immune sites.

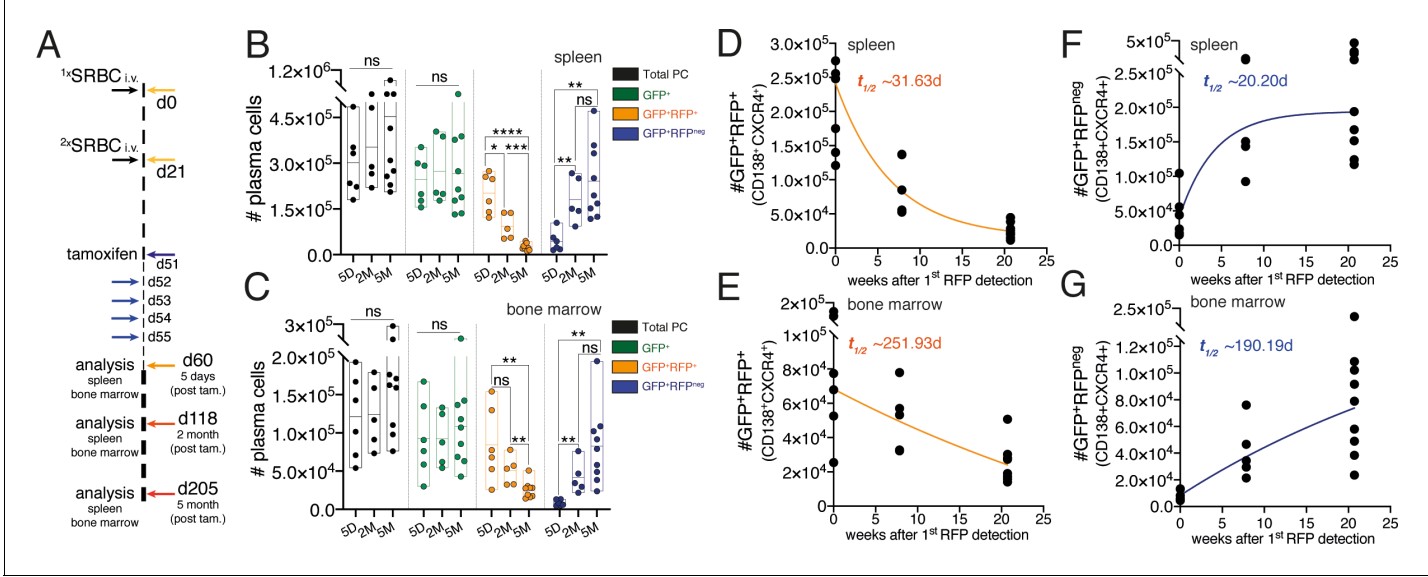

**Figure 6.** Inclusive analysis of plasma cell dynamics reveals tissue-specific homeostatic population turnover. (A) Schematic of the experimental procedure protocol. Mice carrying the *Jchain*[creERT2] and *R26*[lslRFP] alleles were immunized twice with sheep red blood cells (SRBC) intravenously (i.v.) on day 0 and day 21. Mice received tamoxifen treatment for five consecutive days from day 51 to 55 after the first immunization. Spleens and bone marrow of mice were analyzed at day 5, 2-month, and 5-month timepoints after the last tamoxifen administration. (B) Cumulative data for the absolute cell numbers of total PCs, of GFP$^+$ cells (RFP$^+$ and RFP$^{neg}$), GFP$^+$RFP$^+$ cells (cre recombined), and GFP$^+$RFP$^{neg}$ cells (not cre recombined) in the spleen. (C) Cumulative data for the absolute cell numbers of total PCs, of GFP$^+$ cells (RFP$^+$ and RFP$^{neg}$), GFP$^+$RFP$^+$ cells (cre recombined), and GFP$^+$RFP$^{neg}$ cells (not cre recombined) in the bone marrow. (D) Graphical representation of half-life ($t_{1/2}$) of GFP$^+$RFP$^+$CD138$^+$CXCR4$^+$plasma cells in the spleen using the data presented in (B). Graphing of the best-fit curve was performed using the GraphPad Prism eight software. (E) Graphical representation of half-life ($t_{1/2}$) of GFP$^+$RFP$^+$CD138$^+$CXCR4$^+$plasma cells in the bone marrow using the data presented in (C). Graphing of the best-fit curve was performed using the GraphPad Prism eight software. (F) Graphical representation of half-life ($t_{1/2}$) of GFP$^+$RFP$^{neg}$CD138$^+$CXCR4$^+$plasma cells in the spleen using the data presented in (B). Graphing of the best-fit curve was performed using the GraphPad Prism eight software. (G) Graphical representation of half-life ($t_{1/2}$) of GFP$^+$RFP$^{neg}$CD138$^+$CXCR4$^+$plasma cells in the bone marrow using the data presented in (C). Graphing of the best-fit curve was performed using the GraphPad Prism eight software. Each symbol (B-G: 5D n = 6; 2M n = 5; 5M n = 9) represents an individual mouse; small horizontal lines indicate median, minimum, and maximum values. ns = not significant, *=p $\leq$ 0.05 **=p $\leq$ 0.01, ***=p $\leq$ 0.001, ****=p $\leq$ 0.0001 (unpaired Student's *t*-test). Data are representative of three independent experiments (B, C).

The online version of this article includes the following source data for figure 6:

**Source data 1.** Absolute cell numbers of plasma cell populations defined by GFP and RFP expression in the spleen and bone marrow over time.

## Discussion

Plasma cells (PC)s are an essential component of the adaptive immune system. However, compared to other B cell lineage populations the molecular analysis of PC gene function in vivo has lagged behind. One underlying cause is the lack of tools for specific genetic manipulation of PCs, something available for B cells since 1997, more than 20 years ago (*Rickert et al., 1997*), and more recently for B cells at multiple stages of development (*Boross et al., 2009*; *Casola et al., 2006*; *Croker et al., 2004*; *Crouch et al., 2007*; *de Boer et al., 2003*; *Dogan et al., 2009*; *Düber et al., 2009*; *Georgiades et al., 2002*; *Hobeika et al., 2006*; *Kraus et al., 2004*; *Kwon et al., 2008*; *Moriyama et al., 2014*; *Robbiani et al., 2008*; *Schweighoffer et al., 2013*; *Shinnakasu et al., 2016*; *Weber et al., 2019*; *Yasuda et al., 2013*).

Here we used a systematic analysis of PC-associated factors for which expression is increased during PC differentiation (*Xbp1*, *Jchain*, *Scd1*, *Irf4*, and *Prdm1*) with the aim of identifying suitable candidates for the generation of PC-specific tools. *Blimp1*, a critical transcription factor for the establishment of the PC program, displayed an expression pattern unspecific to PCs. This was unsurprising given the knowledge that *Blimp1* is expressed by other hematopoietic and non-hematopoietic cells (*John and Garrett-Sinha, 2009*), including germ cells (*Ohinata et al., 2005*). Among the analyzed factors we found that *Jchain* displayed the highest RNA expression level in PCs and had the most PC-specific expression pattern. We considered *Jchain* to be the most suitable candidate for the generation of PC-specific tools. Next, we searched among alleles generated by the

EUCOMMTools consortium and identified a Jchain^creERT2 allele which through a comprehensive series of experiments we validated for specific genetic manipulation of PCs.

GFP expression as a reporter for *Jchain* was expressed in a very small subset of B cells (<2%, primarily GC B cells) and in most plasma cells. The GFP expressing B cells were mostly negative for BLIMP1 and IRF4 expression. This data agrees with previous work using in vitro cultures of *Blimp1* deficient B cells suggesting that the initiation of *Jchain* expression does not require BLIMP1 (*Kallies et al., 2007*). Thus, the identified population of cells in vivo marked by low GFP expression and high surface expression of B220 (GFP^low^B220^hi^), possibly contains PC precursor cells and future analysis may provide information on the mechanisms underlying the very first steps of PC differentiation. It is important to note, however, that the analysis of the GFP^low^B220^hi^ population requires caution as it contains false GFP positive cells (i.e. background cells). If the underlying reason for the false GFP positivity within the GFP^low^B220^hi^ population is autofluorescence; a common issue when GFP is weakly expressed because autofluorescence typically displays similar excitation and emission characteristics to GFP (*Shapiro, 1995*; approaches that quench autofluorescence may reduce or abrogate background; *Shilova et al., 2017*). Alternatively, the experimenter may design a more stringent gate, possibly increasing specificity and thus reducing contamination as long as the loss of a fraction of lowly expressing GFP cells is acceptable and consideration is taken that such stringency may itself introduce bias in the analysis.

*Jchain*^creERT2^ mediated genetic manipulation was only effective in PCs and data of others deposited in bioRxiv during the revision of this work supports this finding (*Ayala et al., 2020*). In the *Jchain*^creERT2^ allele GFP and cre^ERT2^ are linked by a self-cleaving 2A peptide that ensures a near equitable co-expression of the proteins (*Szymczak et al., 2004*). As consequence, it is likely that the effectiveness of cre recombination correlates with the level of cre^ERT2^ expression because PCs expressed the highest level of *Jchain*, as suggested by GFP expression. Of note, the *Jchain*^creERT2^ allele analyzed in this work retained the puromycin selection cassette that is flanked by rox recombination sites. This knowledge should be taken in consideration in future experiments involving compound mutant mice that make use of dre recombinase (*Anastassiadis et al., 2009*). On occasion it has been observed that retention of the selection cassette reduces gene expression of the targeted locus (*Meyers et al., 1998*; *Nagy et al., 1998*). Thus, inadvertent or intentional removal of the puromycin selection cassette may increase cre^ERT2^ expression, including in B cells, possibly reducing the PC specificity of *Jchain*^creERT2^ mediated genetic manipulation. Finally, in the *Jchain*^creERT2^ allele the expression of *Jchain* is interrupted by a GFP-2A-cre^ERT2^ cassette and induction of cre-mediated recombination deletes *Jchain* exon two that is loxP-flanked. It was previously shown that heterozygous deletion of *Jchain* displayed an intermediate phenotype to that of knockout mice (*Lycke et al., 1999*). This knowledge must be considered when performing *Jchain*^creERT2^ mediated genetic manipulation, including the need to use the *Jchain*^creERT2^ allele without the targeted manipulation as control, and the utility of the *Jchain*^creERT2^ allele in homozygosity. Future iterations of *Jchain*^creERT2^ alleles in which JCHAIN protein expression is preserved should be considered.

It is currently thought that *Jchain* expression occurs in all PCs regardless of their isotype (*Castro and Flajnik, 2014*; *Johansen et al., 2000*; *Mather et al., 1981*). The characterization of the *Jchain*^creERT2^ allele demonstrated at the single cell level that *Jchain* expression indeed occurs in PCs across immunoglobulin isotypes. However, we observed the occurrence of both GFP^+^ PCs and of a smaller fraction of GFP^neg^ PCs, which in a first approximation suggested that *Jchain*^neg^ PCs exist. Nevertheless, it is important to consider that although GFP is driven by the endogenous *Jchain* promoter, it does not directly measure JCHAIN itself at the transcriptional and translational level. Further investigation of GFP^+^ and GFP^neg^ PCs in mice carrying the *Jchain*^creERT2^ allele is required.

Using the *Jchain*^creERT2^ allele we performed inclusive genetic timestamping of PCs, independent of the time at which cells were generated, cell cycle status, and localization. We uncovered that the numbers of total PCs in the spleen and bone marrow remained constant for at least 5 months. Notably, within the GFP^+^ PC population, the decay in numbers of genetically labeled PCs (GFP^+^RFP^+^) was compensated by that of unlabeled PCs (GFP^+^RFP^neg^), supporting that PC turnover is homeostatically regulated in these tissues. Homeostatic control of mature B cell numbers is widely accepted (*Crowley et al., 2009*). In mature B cells, the expression of a B cell receptor is crucial for cell survival (*Srinivasan et al., 2009*; however, BAFF is the limited resource that defines the boundaries of the biological 'space'; *Crowley et al., 2009*; *Srinivasan et al., 2009*). On other hand, the regulation of PC numbers remains unclear. PCs express the BCMA receptor that allows the sensing of both BAFF and

APRIL, and BCMA deficient mice have reduced PC numbers (*O'Connor et al., 2004*). However, because BCMA is expressed in B cells committed to PC differentiation further studies are required to disentangle formation and maintenance (*Mackay et al., 2003*). The *Jchain*creERT2 allele is therefore ideally suited to tackle these questions. PC homeostatic regulation could also be the reflection of a limited number of niches in a given organ which would then limit the number of PCs (*Höfer et al., 2006*; *Khodadadi et al., 2019*; *Lightman et al., 2019*; *Lindquist et al., 2019*; *Wilmore and Allman, 2017*).

In this work, we have not determined if the decay in numbers of genetically labeled PCs (GFP+-RFP+) was the reflection of cell death and/or migration away from the tissue analyzed. A possible strategy to determine the latter would be the use of a photoactivatable fluorescent reporter (*Patterson and Lippincott-Schwartz, 2002*) in combination with the *Jchain*creERT2 allele to label through photoactivation PCs in a specific tissue and investigate their migratory patterns. Knowledge on the mechanisms underlying PC homeostasis and turnover is important as these are directly related to long-term protection from infection, vaccination, and cancer pathogenesis. The *Jchain*creERT2 allele is highly suited to perform these investigations as it allows genetic manipulation of PCs in vivo in their microenvironment, and to retrieve live time-stamped PCs for downstream analysis.

## Materials and methods

### Key resources table

| Reagent type (species) or resource | Designation | Source or reference | Identifiers | Additional information |
|---|---|---|---|---|
| Genetic reagent (*Mus musculus*, C57BL/6) | *Jchain*creERT2. *Jchain*tm1(EGFP/cre/ERT2)Wtsi | Wellcome Trust Sanger Institute (WTSI) | MGI: 5633773 | The allele was purchased from EMMA mouse repository in agreement with WTSI, mice were rederived at the Francis Crick Institute. |
| Antibody | Anti-mouse Blimp1 (host species: rat, clone: 6D3) | BD Biosciences | Cat#: 565002 | FACS (1:100) |
| Antibody | Anti-mouse CD16/32 Fc Block (host species: rat, clone: 2.4G2) | BD Biosciences | Cat#: 553141 | FACS (1:200) |
| Antibody | Anti-mouse CD19 (host species: rat, clone: 1D3) | BD Biosciences | Cat#: 563557 | FACS (1:200) |
| Antibody | Anti-mouse CD23 (host species: rat, clone: B3B4) | BD Biosciences | Cat#: 563986 | FACS (1:200) |
| Antibody | Anti-mouse CD38 (host species: rat, clone: 90) | BD Biosciences | Cat#: 760361 | FACS (1:200) |
| Antibody | Anti-mouse CD95 (host species: hamster, clone: Jo2) | BD Biosciences | Cat#: 562633 | FACS (1:200) |
| Antibody | Anti-mouse IgG1 (host species: rat, clone: A85-1) | BD Biosciences | Cat#: 560089 | FACS (1:200) |
| Antibody | Anti-mouse CD138 (host species: rat, clone: 281–2) | BD Biosciences | Cat#: 740880 | FACS (1:200) |
| Antibody | Anti-mouse B220 (host species: rat, clone: RA3-6B2) | BioLegend | Cat#: 103247 | FACS (1:200) |
| Antibody | Anti-mouse CD11c (host species: hamster, clone: N418) | BioLegend | Cat#: 117333 | FACS (1:200) |
| Antibody | Anti-mouse CD19 (host species: rat, clone: 6D5) | BioLegend | Cat#: 115543 | FACS (1:200) |
| Antibody | Anti-mouse CD21/35 (host species: Rat, clone: 7E9) | BioLegend | Cat#: 123421 | FACS (1:200) |
| Antibody | Anti-mouse CD43 (host species: rat, clone: 1B11) | BioLegend | Cat#: 121223 | FACS (1:200) |
| Antibody | Anti-mouse CD86 (host species: rat, clone: GL1) | BioLegend | Cat#: 105013 | FACS (1:200) |
| Antibody | Anti-mouse BP1 (host species: rat, clone: 6C3) | Ebioscience | Cat#: 13–5891 | FACS (1:200) |

*Continued on next page*

*Continued*

| Reagent type (species) or resource | Designation | Source or reference | Identifiers | Additional information |
|---|---|---|---|---|
| Antibody | Anti-mouse CD5 (host species: rat, clone: 53–7.3) | Ebioscience | Cat#: 13-0051-82 | FACS (1:200) |
| Antibody | Anti-mouse CXCR4 (host species: rat, clone: 2B11) | Ebioscience | Cat#: 46-9991-82 | FACS (1:200) |
| Antibody | Anti-mouse IgA (host species: rat, clone: 11-44-2) | Ebioscience | Cat#: 13–5994 | FACS (1:200) |
| Antibody | Anti-mouse IgM (host species: rat, clone: II/41) | Ebioscience | Cat#: 25–5790 | FACS (1:300) |
| Antibody | Anti-mouse IRF4 (host species: rat, clone: 3E4) | Ebioscience | Cat#: 25-9858-80 | FACS (1:200) |
| Commercial assay or kit | BD Cytofix/CytoPerm Fixation/Permeabilization Kit | BD Biosciences | Cat#: 554714 | |
| Commercial assay or kit | Zombie NIR Fixable Viability Kit | BioLegend | Cat#: 423106 | (1:200) |
| Commercial assay or kit | CD43 (Ly-48) Micro Beads, mouse | Miltenyi Biotec | Cat#: 130-049-801 | |
| Chemical compound, drug | (Z) 4-hydroxytamoxifen | Sigma-Aldrich | CAS: 68047-06-3 | |
| Chemical compound, drug | Tamoxifen | Sigma | Cat#: T5648-5G | |
| Chemical compound, drug | Sunflower seed oil from *Helianthus annuus* | Sigma-Aldrich | Cat#: S5007-250ml | |
| Software, algorithm | FACSDiva software | BD | V9.0 | |
| Software, algorithm | FlowJo | BD | V10 | |
| Software, algorithm | Prism | GraphPad | V7, V8 | |
| Other | Sheep red blood cells (SRBCs) | TCS Biosciences Ltd | Cat#: SB054 | |

## Mice

The *Jchain*$^{creERT2}$ allele was purchased from EMMA in agreement with the Wellcome Trust Sanger Institute (Jchain targeted allele Wtsi MGI:5633773, genebank https://www.i-dcc.org/imits/targ_rep/alleles/43805/escell-clone-genbank-file) and mice were rederived at the The Francis Crick Institute. The allele contains a splice acceptor site (SA), an *EGFP-2A-creERT2* expression cassette and a poly-A tail in the intron between exons 1 and 2 under the *Jchain* promoter. In addition, exon 2 is loxP-flanked and the allele also contains a rox-flanked puromycin resistance cassette. These mice were crossed to carry a *Rosa26*$^{lslRFP}$ cre recombination reporter allele (*R26*$^{lsl.RFP}$) allele that expresses a non-toxic tandem-dimer red fluorescent protein upon cre-mediated deletion of a floxed STOP cassette (*Luche et al., 2007*). Mice were maintained on the C57BL/6 background and bred at The Francis Crick Institute biological resources facility under specific pathogen-free conditions. Animal experiments were carried out in accordance with national and institutional guidelines for animal care and were approved by The Francis Crick Institute biological resources facility strategic oversight committee (incorporating the Animal Welfare and Ethical Review Body) and by the Home Office, UK. All animal care and procedures followed guidelines of the UK Home Office according to the Animals (Scientific Procedures) Act 1986 and were approved by Biological Research Facility at the Francis Crick Institute. The age of mice ranged between 15–30 weeks as specified.

## Immunization and in vivo induction of cre activity

Mice were injected intravenously with $1 \times 10^9$ sheep red blood cells (SRBCs, TCS Biosciences Ltd) in PBS. For the induction of cre activity, 4 mg tamoxifen (SIGMA T5648) dissolved in sunflower seed oil

were administered by oral gavage to mice once per day for multiple days depending on experimental design.

### In vitro B cell culture and induction of cre activity

Splenic cells were harvested, and B cells were isolated using CD43 (Ly-48) MicroBeads, mouse (Miltenyi Biotec). The purity of B cells was determined by flow-cytometry (>95%). B cells were cultured in 96-well round-bottom plates (Falcon) in B cell media (DMEM high glucose/Glutamax from Thermo Fisher Scientific supplemented with 10% fetal bovine serum F7524 from Sigma, 100 U/mL Penicillin, 100 µg/mL Streptomycin from Life Technologies, 10 mM HEPES buffer solution from Life Technologies, 100 µM MEM non-essential amino acids from Thermo Fisher Scientific, 1 mM sodium pyruvate from Life Technologies, and 50 uM β-mercaptoethanol from Sigma) at 1 million/mL concentration (200,000 cells/well) with 10 ug/mL LPS and varied concentrations of (Z) 4-hydroxytamoxifen (Sigma-Aldrich). Analysis was performed using flow-cytometry at 48, 72, or 96 hr of culture.

### Flow cytometry

Single cell suspensions were stained with antibodies. We used Zombie NIR Fixable Viability Kit (BioLegend) for live/dead discrimination. For intracellular staining, we fixed cells using the BD CytoFix/Cytoperm (BD Biosciences) kit as per manufacturer instructions. Samples were acquired on a BD LSRFortessa analyzer using FACSDiva software (BD) and analyzed on FlowJo software.

### Quantification and statistical analysis

Data were analyzed with unpaired two-tailed Student's $t$-test; a p-value=$p \leq 0.05$ was considered significant. Prism (v7 and v8, GraphPad) was used for statistical analysis. A single asterisk ($^*$) in the graphs of figures represents a p-value$\leq 0.05$, double asterisks ($^{**}$) a p-value$\leq 0.01$, triple asterisks ($^{***}$) a p-value$\leq 0.001$, quadruple asterisks ($^{****}$) a p-value$\leq 0.0001$, and 'ns' stands for not statistically significant (i.e. a p-value>0.05). Nonlinear regression (curve fit) using (v8, GraphPad) was used to calculate half-life ($t_{1/2}$) of the population and followed a one-phase decay model, with no special handling of outliers, robust regression and strict convergence criteria and no weighting with a 1000 maximum number of iterations.

## Acknowledgements

We thank the members of the Immunity and Cancer laboratory (FCI, London) for critical discussions and review of the manuscript; the FCI BRF and Flow-cytometry platforms for expert advice and technical support. This work was supported by The Francis Crick Institute which receives its core funding from Cancer Research UK (FC001057), the UK Medical Research Council (FC001057), the Wellcome Trust (FC001057) to DPC, by CRUK [C355/A26819], FC AECC [C355/A26819], AIRC [C355/A26819] under the Accelerator Award Program to DPC, and an MRC career development award MR/J008060/1 to DPC.

## Additional information

### Funding

| Funder | Grant reference number | Author |
| --- | --- | --- |
| Cancer Research UK | FC001057 | Dinis Pedro Calado |
| Medical Research Council | FC001057 | Dinis Pedro Calado |
| Wellcome Trust | FC001057 | Dinis Pedro Calado |
| Cancer Research UK | [C355/A26819] | Dinis Pedro Calado |
| FC AECC | [C355/A26819] | Dinis Pedro Calado |
| AIRC | [C355/A26819] | Dinis Pedro Calado |
| Medical Research Council | MR/J008060/1 | Dinis Pedro Calado |

The funders had no role in study design, data collection and interpretation, or the decision to submit the work for publication.

### Author contributions
An Qi Xu, Conceptualization, Formal analysis, Investigation, Methodology, Writing - original draft, Writing - review and editing; Rita R Barbosa, Conceptualization, Investigation, Methodology; Dinis Pedro Calado, Conceptualization, Resources, Data curation, Formal analysis, Supervision, Funding acquisition, Investigation, Methodology, Writing - original draft, Project administration, Writing - review and editing

### Author ORCIDs
An Qi Xu (iD) https://orcid.org/0000-0001-8008-6462
Dinis Pedro Calado (iD) https://orcid.org/0000-0001-8239-7184

### Ethics
Animal experimentation: Animal experiments were carried out in accordance with national and institutional guidelines for animal care and were approved by The Francis Crick Institute biological resources facility strategic oversight committee (incorporating the Animal Welfare and Ethical Review Body) and by the Home Office, UK licence number PCE886633. All animal care and procedures followed guidelines of the UK Home Office according to the Animals (Scientific Procedures) Act 1986 and were approved by Biological Research Facility at the Francis Crick Institute.

### Decision letter and Author response
Decision letter https://doi.org/10.7554/eLife.59850.sa1
Author response https://doi.org/10.7554/eLife.59850.sa2

## Additional files

### Supplementary files
• Transparent reporting form

### Data availability
All data generated or analysed during this study are included in the manuscript and supporting files.

The following previously published dataset was used:

| Author(s) | Year | Dataset title | Dataset URL | Database and Identifier |
|---|---|---|---|---|
| Heng TS, Painter MW, Immunological Genome Project Consortium | 2008 | ImmGen ULI RNA-seq data | https://www.ncbi.nlm.nih.gov/geo/query/acc.cgi?acc=GSE127267 | NCBI Gene Expression Omnibus, GSE127267 |

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
