## [Decision Letter]

**Acceptance summary:**

The generation and analysis of *Jchain*^creERT2^ tool described in this manuscript will be a valuable addition to the arsenal of regulatable Cre transgenes available to probe the immune system. Your studies have already provided new insights into plasma cell homeostasis that will stage for the use of this mouse strain for additional analyses. We wish you continued success in probing the biology of plasma cells via development and use of novel reagents such as the one described in this manuscript.

**Decision letter after peer review:**

Thank you for submitting your article "Genetic timestamping of plasma cells in vivo reveals homeostatic population turnover" for consideration by *eLife*. Your article has been reviewed by two peer reviewers, and the evaluation has been overseen by a Reviewing Editor and Satyajit Rath as the Senior Editor. The reviewers have opted to remain anonymous.

The reviewers have discussed the reviews with one another and the Reviewing Editor has drafted this decision to help you prepare a revised submission.

Summary:

Both reviewers found this work to be important and characterization of the use of *Jchain*^creERT2^ mice as a valuable resource for inducible manipulation of genes in plasma cells (PC). The reviewers noted that "The present manuscript presents experiments that convincingly demonstrate the suitability of a *Jchain*^creERT2^ mice (inducible Cre) to delete loxP-flanked genes specifically in the plasma cell (PC) lineage in an inducible fashion" and the work "addresses an important question in plasma cell biology by developing a genetic time-stamping in vivo tool to look at plasma cell turnover". However, both reviewers raised points enumerated below that need to be addressed before further consideration of this manuscript for publication.

Essential revisions:

1) Account for the presence of GFP^lo^B220^hi^ cells that appear to be present in control mice as well. Do these represent non-specific staining? If so, how can the contribution of GFP from the modified J chain allele be distinguished from the background? This question is pertinent to Figures 1C, 2C and 3B and could be resolved by providing staining in the GFP^lo^B220^hi^ gate for CD138, CXCR4, Blimp1 and IRF4.

2) Clarify the basis of the conclusion that the "…analysis did not support the concept that the J chain is expressed by all PCs".

3) Modify the fourth paragraph of the Discussion, there is sentence in the middle which appears to have some missing words.

4) Quote the plasma cell paper from Singh/Staudt labs ion the role of IRF4 (Sciammas et al., 2006).

5) Justify the interpretation critical for the analysis of homeostatic turnover of PC that decline in RFP^+^ cells is due to loss of PC rather than other possibilities such as migration of PCs to other niches. While this would ideally involve additional experiments with BrdU-labeled cells, a strong rationale for their interpretation along with a succinct discussion of possible limitations in their analysis would suffice for reconsideration of the manuscript.

6) As an extension to the above point, were RFP^+^ cells scored in other PC survival niches?

---

## [Author Response]

Essential revisions:1) Account for the presence of GFP^lo^B220^hi^ cells that appear to be present in control mice as well. Do these represent non-specific staining? If so, how can the contribution of GFP from the modified J chain allele be distinguished from the background? This question is pertinent to Figures 1C, 2C and 3B and could be resolved by providing staining in the GFP^lo^B220^hi^ gate for CD138, CXCR4, Blimp1 and IRF4.

This is valid comment. We defined the GFP^low^B220^hi^ gate in order to not disregard the occurrence of cells with a low-level of GFP expression in mice carrying the *Jchain-cre^ERT2^* allele. However, as the reviewer’s point out this led to the inclusion of a small fraction of false GFP positive cells as indicated by the analysis of control mice (wild-type C57BL/6) that are GFP negative. Following the reviewer’s comment, we analyzed data to quantify the contribution of GFP from the *Jchain-cre^ERT2^* allele within the GFP^low^B220^hi^ gate. We found that the fraction of cells within the GFP^lo^B220^hi^ gate was significantly increased in mice carrying the *Jchain-cre^ERT2^* allele (Figure 2—figure supplement 1A). The cells within the GFP^low^B220^hi^ gate of mice carrying the *Jchain-cre^ERT2^* allele were enriched for B cells (CD138^neg^B220^hi^CD19^+^) compared to control mice and these B cells mostly represented germinal center B cells (Figure 2—figure supplement 1B) in agreement to the data presented in Figure 2F and G of the manuscript. Thus, although the GFP^low^B220^hi^ gate includes 0 to 0.16% of GFP negative cells as determined in control mice, it provides a significant enrichment for GFP^low^ cells in mice carrying the *Jchain-cre^ERT2^* allele.

With respect to CD138^+^, CD138^+^CXCR4^+^ and BLIMP1^+^IRF4^+^ cells within the GFP^low^B220^hi^ population, these were enriched in a fraction of mice carrying the *Jchain-cre^ERT2^* allele, approximately 30%, compared to control mice (Figure 2—figure supplement 1C-D). However, for 70% of mice carrying the *Jchain-cre^ERT2^* allele the GFP^low^B220^hi^ population was not enriched for cells with these markers compared to control background.

In summary, whereas the experimental analysis of the GFP^low^B220^hi^ gate determines the enrichment for cells expressing a low-level GFP in mice carrying the *Jchain-cre^ERT2^* allele, representing mostly germinal center B cells, caution should be taken when using this gating strategy to study enrichment for cells with plasma cell markers, as this gate may contain an unacceptable level of contamination by non-GFP positive cells. If the underlying reason for the false GFP positivity within the GFP^low^B220^hi^ population is autofluorescence; a common issue when GFP is weakly expressed because autofluorescence typically displays similar excitation and emission characteristics to GFP (Shapiro, 1995); approaches that quench autofluorescence may reduce or abrogate background (Shilova et al., 2017). Alternatively, the experimenter may design a more stringent gate, possibly increasing specificity and thus reducing contamination as long as the loss of a fraction of lowly expressing GFP cells is acceptable and consideration is taken into account that such stringency could also introduce bias in the analysis.

We have included this data as a Figure 2—figure supplement 1 and Figure 3—figure supplement 1 and present and discuss the data in the Results and Discussion sections.

2) Clarify the basis of the conclusion that the "…analysis did not support the concept that the J chain is expressed by all PCs".

Following the reviewer’s comment we considered that this conclusion is not accurate. The analysis of mice carrying the *Jchain-cre^ERT2^* allele revealed that roughly 60-90% of CD138^+^CXCR4^+^ cells express GFP (Figure 2—figure supplement 2). As GFP expression is driven by the endogenous *Jchain* promoter this data may suggest in a first approximation that a minor fraction of plasma cells does not express JCHAIN. However, there are inherent limitations of using GFP as a reporter as it does not directly measure JCHAIN itself at the transcriptional and translational level.

We have corrected our statement and added these considerations in the manuscript: Discussion section, fifth paragraph.

3) Modify the fourth paragraph of the Discussion, there is sentence in the middle 12 which appears to have some missing words.

We thank the reviewers for highlighting that correction is needed for this sentence. We now wrote in Discussion section: “Thus, inadvertent or intentional removal of the puromycin selection cassette may increase cre^ERT2^ expression, including in B cells, possibly reducing the PC specificity of *Jchain^creERT2^* mediated genetic manipulation.”.

4) Quote the plasma cell paper from Singh/Staudt labs ion the role of IRF4 (Sciammas et al., 2006).

We apologize for this omission. The reference as now been added when introducing IRF4 in the first paragraph of the Introduction, and in the first paragraph of the Results subsection “*Jchain* expression correlates with that of IRF4 and BLIMP1”, where the analysis of IRF4 is described.

5) Justify the interpretation critical for the analysis of homeostatic turnover of PC that decline in RFP^+^ cells is due to loss of PC rather than other possibilities such as migration of PCs to other niches. While this would ideally involve additional experiments with BrdU-labeled cells, a strong rationale for their interpretation along with a succinct discussion of possible limitations in their analysis would suffice for reconsideration of the manuscript.

This is a relevant comment. We agree that the cell number turnover of labelled RFP^+^ could be at least in part due to the migration of cells out of the tissues analyzed. Thus, to be clear, we agree that the data presented in the manuscript does not address if the decay in RFP^+^ plasma cell numbers is due to cell death and/or migration.

We were, however, struck by the observation that the absolute number of total plasma cells in spleen and bone marrow remained largely unaltered over time (5 month of analysis), suggesting that the number of plasma cells is homeostatically regulated in those tissues. This observation in addition to the finding that the loss in cell numbers of RFP^+^GFP^+^ plasma cells over-time was “compensated” by increased cell numbers of RFP^neg^GFP^+^ plasma cells, supported homeostatic population turnover in the tissues analyzed. As mentioned above it remains to be determined whether the turnover of RFP^+^GFP^+^ plasma cells is due to cell death and/or cell migration. It is also not possible to formulate at this stage a conclusion at the organism level, that would require an in-depth analysis of multiple tissues of the body.

To reflect these considerations, we included in the title of the manuscript the term “tissue-specific” and modified the text of the manuscript highlighting the limitations of the analysis: Discussion section, last paragraph.

We have considered the proposition to perform BrdU cell labelling to address the question whether cell migration occurs between tissues. BrdU labelling would indeed allow the identification of nascent plasma cells generated in secondary lymphoid organs namely spleen, lymph nodes and gut associated lymphoid tissue, the major sites of B-to-plasma cell differentiation. This would permit the investigation of the migration of nascent RFP^+^ plasma cells into the bone marrow and other sites where plasma cells are not thought to be formed. However, the calculation of the re-circulation of bone marrow RFP^+^BrdU^+^ plasma cells into secondary lymphoid organs would likely be confounded by the presence of RFP^+^ plasma cells labelled with BrdU in situ or derived from other tissues. The follow up of the migration of already formed quiescent RFP^+^BrdU^neg^ plasma cells at the time of BrdU labelling would not be possible.

A possible strategy to overcome these limitations would be the use of a photoactivatable fluorescent reporter in combination with the *Jchain-cre^ERT2^* allele i.e. to label through photoactivation plasma cells in a specific tissue and investigate their migratory patterns.

We comment on this strategy in the Discussion section of the manuscript: last paragraph.

6) As an extension to the above point, were RFP^+^ cells scored in other PC survival niches?

We have not determined the dynamics of RFP labelled plasma cells in other tissues besides spleen and bone marrow. However, we agree with the reviewers on the relevance of investigating other potential plasma cell niches, including gut and associated lymphoid tissues.